# Effectiveness and challenges of digital tools implementation for enhancing infectious disease surveillance data quality in low- and middle-income countries: A systematic review protocol

Oluwatosin Olu-Abiodun[1], Aderinsola Faturoti[2], Akinmade Adepoju [iD][3], Davies Adeloye[4], Akindele Adebiyi[5], Olumide Abiodun [iD][6]*

1 Department of Nursing, Crescent University, Abeokuta, Ogun State, Nigeria, 2 Department of Community Medicine, Babcock University Teaching Hospital, Ilishan-Remo, Ogun State, Nigeria, 3 Department of Community Medicine, Babcock University Teaching Hospital, Ilishan-Remo, Ogun State, Nigeria, 4 School of Health and Life Sciences, Nursing &Midwifery, Teesside University, United Kingdom, 5 Department of Community Medicine, University of Ibadan, Oyo State, Nigeria, 6 Department of Community Medicine, Babcock University Teaching Hospital, Ilishan-Remo, Ogun State, Nigeria

* olumiabiodun@gmail.com

## Abstract

Prospero ID-1023840

## Background

Monitoring infectious diseases is essential for preventing and controlling outbreaks, especially in low- and middle-income countries (LMICs), where issues like poor infrastructure, lack of trained staff, and limited resources can make data collection challenging. Digital tools such as mobile health apps and electronic reporting systems show promise in addressing these problems. However, it's still unclear how well these tools actually improve the quality of data, like how quickly information is reported, how accurate it is, whether all necessary data is captured, and if the data can be trusted.

## Objectives

This review aims to explore three main points: (1) how digital tools influence the quality of infectious disease data in LMICs; (2) what factors help or hinder their successful use; and (3) what recommendations can be made for policymakers and health workers based on the evidence.

## Methods

We will search several databases, including PubMed/MEDLINE, EMBASE, Scopus, CINAHL, and Google Scholar, for studies published from January 2000 to July 2025.

**Data availability statement:** No datasets were generated or analysed during the current study. All relevant data from this study will be made available upon study completion.

**Funding:** The author(s) received no specific funding for this work.

**Competing interests:** The authors have declared that no competing interests exist.

To further reduce publication bias, we will search the following institutional repositories (African Health Observatory and Indian Council of Medical Research). The types of studies are randomised trials, quasi-experimental studies, and mixed-methods evaluations that compare digital solutions with traditional methods in LMIC settings. Data extracted will include outcomes such as delays in reporting, error rates, and completeness, and factors like infrastructure and workforce readiness. The quality of each study will be assessed using ROBINS-I for non-randomized studies and ROB2 for randomized controlled trials. Where possible, we will combine data statistically using meta-analysis and analyse qualitative findings for deeper insights.

## Expected Outcomes

This review will offer a clear picture of how effective digital tools are in improving disease surveillance. It will identify common challenges, such as poor connectivity and issues with system integration, and emphasize factors that lead to success, like proper training and government support. Overall, the findings will help shape better strategies to strengthen digital disease monitoring, finally contributing to stronger global health security.

## Introduction

Infectious disease surveillance is at the heart of public health practice as it facilitates the prevention, early detection, and control of outbreaks [1]. High-quality surveillance data are critical for mounting timely and effective outbreak responses, especially in Low- and Middle-Income Countries (LMICs), where poor infrastructure, workforce shortages, and limited financial resources pose significant constraints to the traditional surveillance systems. The emergence and increasing adoption of digital surveillance tools to enhance data collection, reporting, and analysis for optimizing data quality, early disease outbreak detection, and overall disease control efforts offer a potent solution to these challenges. Despite the significant promise held by these digital tools, their effectiveness in enhancing the quality of infectious disease surveillance data remains unclear [2], necessitating a critical and systematic inquiry.

In LMICs, where the health systems are weak and the burden of infectious diseases is disproportionately high, surveillance data suffer quality challenges relating to timeliness, accuracy, completeness, and reliability. These issues exert a significant negative effect on decision-making and public health response to outbreaks. The need for investments in comprehensive surveillance systems is, therefore, evident to enhance global health security and prevent outbreaks [3].

The last two decades have been characterized by integrating digital technology into health care, which has led to several digital health platforms for strengthening infectious disease surveillance [1]. Tools like electronic health records (EHRs), mobile health (mHealth) applications, geographic information systems (GIS), and artificial intelligence (AI)-driven analytics, internet-based participatory surveillance, and

cloud-based reporting platforms offer the potential to improve data quality by facilitating real-time reporting, automating data validation, ensuring seamless data integration, and enhancing data accessibility. One such tool, the District Health Information Software 2 (DHIS2), is an open-source platform with successful adoption in over 75 LMICs for strengthening health information systems [4].

The tools are grounded in public health informatics, a discipline that integrates epidemiologic methods into information technology to optimize disease surveillance. Several frameworks provide theoretical underpinnings for the deployment of digital surveillance tools. The Health Information System (HIS) framework highlights the system-wide factors that influence data quality, particularly infrastructure, governance, and interoperability. Conversely, the Technology Acceptance Model (TAM) focuses on the end users ' acceptance of digital tools as it relates to their perception of its usefulness and ease of use. Furthermore, the Information Cycle Model (ICM) highlights processes like data collection, processing, and dissemination. ICM underlines the point at which data tools can potentially improve data quality [5].

Despite the huge potential for improved surveillance data quality, several factors mitigate the effectiveness of digital tools in LMICs. Widespread infrastructural and connectivity technology deficits impede digital tool functionality and cause transmission delays, recurrent system downtimes, and server failures, which in turn result in data incompleteness or obsoleteness and suboptimal real-time surveillance potential [6]. The absence of standardized reporting protocols and system interoperability is another issue that hinders the smooth flow of data at regional, national, and international surveillance networks [4]. Also, the absence of strong regulatory frameworks increases the susceptibility to data breaches and unauthorized access, thereby impinging on the confidentiality of critical health data [5]. Widespread shortages of trained personnel, heightened by the prevailing brain drain syndrome, and low digital literacy within the existing health workforce provide the recipe for reporting errors and data inconsistencies [2]. Furthermore, prevailing sociocultural beliefs and misconceptions in many LMICs about disease reporting limit data accuracy and create ethical dilemmas in the implementation of disease surveillance systems [3].

The current evidence on the effectiveness of infectious disease digital surveillance tools for improving data quality in LMICs is mixed. Some studies reported improved timeliness and completeness with automated reporting systems and data collection with mobile devices. For example, a Kenyan study demonstrated a 60% reduction in the reporting of delayed malaria case notification by an SMS-based reporting system [2]. Likewise, an electronic surveillance system improved tuberculosis data accuracy and completeness in India [4]. However, other scholars have demonstrated the persistence of challenges in the implementation of digital surveillance systems. For example, the polio digital reporting system in Nigeria was fraught with data incompleteness because of poor internet connectivity and non-adherence to reporting guidelines by health workers [3]. Furthermore, whereas digital tools enhanced real-time reporting in Uganda, there was persistent non-integration with laboratory data, resulting in suboptimal effectiveness of infectious disease surveillance [5].

Considering the divergence in the evidence, a systematic evaluation is necessary to interrogate the effectiveness of digital surveillance platforms on infectious disease data quality in LMICs. This systematic review will consolidate available evidence to determine the overall impact of digital surveillance tools on infectious disease data quality while also identifying the contextual factors that influence the successful and unsuccessful implementation of digital surveillance. Furthermore, the review will provide policy and strategy recommendations to optimize digital surveillance for infectious diseases in LMICs.

## Materials and methods

### Aims

To identify and critically evaluate peer-reviewed studies published between January 2000 and July 2025 assessing the effectiveness of digital surveillance platforms in improving infectious disease data quality in low- and middle-income countries (LMICs).

The following specific objectives will guide this:

1. To quantitatively and qualitatively assess the impacts of digital surveillance tools compared to conventional surveillance methods on data quality dimensions: timeliness, accuracy, completeness, and reliability.

2. To explore contextual and implementation factors (e.g., infrastructure, workforce capacity, data security, interoperability) influencing the effectiveness of digital surveillance tools.

3. To provide evidence-based recommendations for public health professionals, researchers, and policymakers on deploying digital tools in infectious disease surveillance.

## Methods of the review

**Criteria for selecting studies for this review. Study designs:** Eligible studies will include randomised controlled trials, quasi-experimental designs, cohort and case-control studies, cross-sectional analyses, program evaluations, and mixed-methods studies explicitly evaluating digital surveillance interventions.

**Case definitions:** Studies must clearly define infectious diseases monitored, adhering to internationally recognised or national surveillance case definitions.

**Participants:** Our study participants are health systems, healthcare facilities, or public health agencies operating within low—and middle-income countries (LMICs) that use or assess surveillance systems for detecting and responding to infectious diseases. LMICs will be defined according to the World Bank classification (countries with a gross national income (GNI) per capita of $13,845 or less) categorized using the Atlas method [7].

**Interventions:** These include digital tools or platforms such as mobile applications, electronic reporting systems, web-based dashboards, and integrated health information technologies.

**Comparators:** Encompasses traditional surveillance methods (such as paper-based or non-digital approaches), no surveillance system, or pre-intervention periods in before-and-after studies (comparing baseline data with data collected following digital tool implementation).

**Outcomes:** Primary outcomes include timeliness, accuracy, completeness, and reliability of data. Secondary outcomes cover user acceptability, usability, adoption rates, implementation costs, and feasibility metrics.

Inclusion and Exclusion Criteria

To ensure pertinence, the following criteria were implemented:

Inclusion Criteria

a. Studies that assess the effectiveness of digital surveillance tools in infectious disease surveillance

b. Studies that detail data quality indicators (accuracy, timeliness, completeness, reliability)

c. Studies conducted in LMICs as categorized by the World Bank

d. Randomized controlled trials, observational studies, qualitative studies, and systematic reviews

e. Peer-reviewed articles published from 2000 to 2025

Exclusion Criteria

a. Studies not connected to infectious disease surveillance

b. Studies concentrating solely on non-communicable diseases (NCDs)

c. Studies lacking measurable data quality indicators

d. Conference abstracts, commentaries, and opinion pieces without empirical data

**Search strategy for the identification of studies.** A systematic search will be conducted in five electronic databases: PubMed/MEDLINE, EMBASE, Scopus, and CINAHL, complemented by searches in Google Scholar for relevant grey literature. The search strategy will combine keywords and Medical Subject Headings (MeSH) related to digital surveillance tools (e.g., "mobile health", "digital reporting systems", "electronic surveillance"), infectious diseases (e.g., "malaria", "cholera", "COVID-19"), and LMIC settings (country-specific terms alongside "low-income" or "middle-income"). Searches will be restricted to studies published between January 2000 and July 2025. This is because many of the major advances in digital health technologies, such as mobile health, electronic monitoring platforms, and spatial data systems, began to grow and become widespread around that time, especially in low- and middle-income countries. Also, global efforts to improve disease surveillance, like the World Health Organization's revised framework called the Integrated Disease Surveillance and Response (IDSR), have mostly been developed after 2000 [8]. Because of this, studies published before 2000 are less likely to include information about the modern tools, infrastructure, and systems used for digital disease monitoring in these settings [9].

Reference lists of included studies and relevant systematic reviews will also be screened. There will be no language restriction. Studies in other languages will be translated into English using Google Translate. This is particularly vital given that exclusion based on language is recognised as a source of in systematic reviews, especially within the fields of public health and epidemiology [10,11]. Detailed search strategy is provided in Appendix 1

## Data collection and analysis

**Selection of studies.** Study selection will be conducted in two stages: first, a screening of titles and abstracts to identify potentially eligible studies; and second, a full-text screening of those studies retained after the first stage. Since we expect the search to return a significant number of citations, an initial titles and abstracts screening will be undertaken to exclude irrelevant studies. All the remaining studies will have their titles and abstracts independently assessed for eligibility by two authors (AA and FA). The full articles of the abstracts and titles for which eligibility is not agreed on or remains unclear will be retrieved and jointly examined by both assessors (AA and FA) in conjunction with AO to engender a consensus, as required. Our study will maintain a record of all excluded studies and the reasons for exclusion. The EndNote X8.2 (Bld 11343) software will be used to manage the records of the retrieved studies. Duplicate citations will be automatically detected and eliminated through the deduplication feature in EndNote X8.2. Following this, a manual review will be conducted to verify that no duplicates remain and to confirm the thoroughness of the process.

## Data extraction and management

**Data extraction.** Two reviewers will independently extract data using a standardised extraction form capturing bibliographic details, study design and duration, participant characteristics, intervention specifics, comparator methods, outcome definitions and measures, contextual implementation factors, and quantitative and qualitative outcomes. Discrepancies will be resolved through consensus or arbitration by a third reviewer. Study authors will be contacted through email to provide unavailable data.

**Study characteristics.** For each study, we will document bibliographic details (author, publication year, title, source) country of study, study design (cross-sectional, cohort, randomized controlled trial, mixed methods), study setting (hospital-based, community-based, national surveillance system), and target population (healthcare workers, surveillance officers, general population). The study will also document the study objectives and questions, sample size, and eligibility criteria. The proposed data extraction form is provided in Appendix 2.

**Surveillance tools and intervention details.** To ensure a wholesome assessment of the digital surveillance interventions, the study will extract data on the nature of the tools/platforms (HER, mH apps, GIS-based platforms, AI-driven analytics, DHIS2), tool functionalities, intervention duration, and report of comparison with traditional surveillance approaches.

**Outcomes and data quality indicators.** The study will extract data on the primary outcomes detailed above and their measurement methods. Secondary data to be extracted are the impact on disease detection, outbreak response efficiency, and intervention integration within the broader health information systems.

**Implementation and contextual factors.** Context is critical to the effectiveness of surveillance; therefore, it is vital to evaluate contextual facilitators and barriers to digital tool implementation for disease surveillance. To ensure the foregoing, we will extract data on infrastructure and connectivity challenges, workforce capacity and training, regulatory and ethical considerations, interoperability with existing HIS, and issues with adoption and compliance among users.

**Key findings and effectiveness.** The primary findings of each study will be documented with emphasis on the effect of the digital tool on the quality of surveillance data.

**Risk of bias and quality appraisal.** The Risk of Bias in Non-randomised Studies of Interventions (ROBINS-I) will be used for non-randomized studies [12], while the Cochrane Risk of Bias (RoB 2) tool will be used for randomised controlled trials [13]. For qualitative and mixed-methods studies, we will use the Critical Appraisal Skills Programme (CASP) to assess each study's credibility, relevance, and methodological rigour [14].

We will utilise ROBINS-I for quality appraisal of non-randomized studies, and tailor it to different study designs. For cohort studies, all seven domains remain relevant, with particular emphasis on confounding and selection bias due to the importance of temporal relationships. In contrast, for cross-sectional studies, issues related to confounding, participant selection, and outcome measurement continue to be pertinent; however, domains concerning intervention classification and deviations from the planned interventions may be less applicable or may require reinterpretation based on how exposure is defined. Customising each domain involves explicitly defining what constitutes the 'intervention' as the key exposure and contextualising bias assessments within the specific temporal and structural aspects characteristic of each study design. The assessment will be conducted by two independent assessors, while discrepancies will be resolved through a consensus-building process that may involve a third assessor. Studies will be categorised by risk of bias as low, moderate, or high.

**Policy and practice implications.** To capture the actionable insights from each study, we will extract data on proposed recommendations from the studies, especially as they relate to researchers, practitioners, and policymakers.

**References and citation information.** Finally, all studies will be referenced for accuracy and reproducibility by documenting their DOIs or PubMed IDs (PMID) using the American Psychological Association (APA) 7th edition referencing style.

## Data synthesis and analysis

The approach to the synthesis of findings from the included studies will be systematic and structured using quantitative and qualitative methods to facilitate a robust evaluation of the effectiveness of digital surveillance tools and platforms on the quality of infectious disease data in LMICs.

Quantitative synthesis will involve random-effects meta-analysis when comparable data are sufficient. Heterogeneity will be assessed using the I² statistic, with subgroup and meta-regression analyses exploring variability sources. Qualitative thematic analysis will involve using the NVivo software to investigate implementation factors affecting tool effectiveness. Mixed-methods integration will deploy a convergent synthesis approach to triangulate quantitative and qualitative findings, providing comprehensive insights. All analyses will adhere to PRISMA 2020 guidelines [15]. The final review report will incorporate a PRISMA 2020 flow diagram, providing a clear visual overview of the study selection process. This diagram will illustrate the number of records identified, screened, excluded, and included at each respective stage to promote better understanding.

## Qualitative synthesis

We will use narrative synthesis to summarise the study findings relating to study characteristics, typology of digital surveillance tools, reports of improvements in surveillance data quality, and the contextual facilitators and barriers of digital

surveillance implementation. These data will be tabulated to ease comparison and underscore the common themes, gaps, and inconsistencies.

For the included studies that use the qualitative methods, we will utilise thematic analysis to tease out the key themes, including the perceived benefit (improvement in outbreak detection and real-time reporting), barriers (infrastructure, resistance to tool adoption, concerns with data privacy), and facilitators (government support, associated training programmes, and interoperability).

However, for the studies combining qualitative and quantitative components, a convergent synthesis method will be used to synthesise the findings.

## Framework for the qualitative evaluation of the effect of digital surveillance tools on data quality

In addition to our quantitative analysis, we'll perform a qualitative review to better understand how digital surveillance tools impact key aspects of data quality, specifically focusing on timeliness, accuracy, completeness, and reliability. This approach will help us interpret findings across different studies, drawing on concepts from digital health evaluation, health information systems, and data governance in low- and middle-income countries (LMICs).

**Conceptual basis.** This evaluation framework is built on well-known methods and guidelines that help us check the quality of data in health information systems, especially in low- and middle-income countries. We mainly rely on the WHO Data Quality Review (DQR) Toolkit, which offers straightforward ways to evaluate aspects like how complete, timely, and accurate the data is [16]. We also look at the Performance of Routine Information System Management (PRISM) Framework, which provides tools to assess how well routine health data systems are working, focusing on both data quality and how data is used [17]. Also, the mHealth Evidence Reporting and Assessment (mERA) Checklist guides us in reporting and reviewing mobile health projects, making sure data quality stays a top priority in digital health efforts [18].

**Core dimensions and operational definitions.** Table 1 summarises the core dimension and operational definition of the data quality indicators that will be assessed in this review.

**Analytic strategy.** The qualitative synthesis will use a framework analysis approach, where we map findings from the studies included onto four key domains. This process involves extracting qualitative insights and discussion points from these studies, followed by coding and organising the data using the four core domains as deductive categories. Besides, we will apply inductive sub-theming to identify unanticipated factors or mechanisms customised to specific contexts, such as infrastructure challenges or user training needs. To further understand and compare the effects across different settings and tools, a narrative synthesis will be conducted to assess the presence, strength, and variability of outcomes.

**Integration with quantitative results.** The insights gained from qualitative analysis will be combined with the quantitative data to offer a more comprehensive understanding of how digital surveillance tools influence data quality. This approach will help interpret the statistical findings, especially in cases where the results are inconclusive or show mixed effects. It will also clarify how these tools work in practice, identifying what helps or hinders their effectiveness, and

**Table 1. Core dimensions and operational definitions of data quality indicators.**

| Dimension | Definition | Indicator/Analysis Focus |
|---|---|---|
| Timeliness | This refers to how quickly data is collected, reported, and made available for decision-making purposes. | Includes aspects like report submission delays, real-time reporting capabilities, and the impact of automation. |
| Accuracy | This measures how closely reported data reflect the actual values or observations. | Key indicators are error rates, discrepancies found during verification, validation checks, and mechanisms like double-entry to ensure correctness. |
| Completeness | This assesses whether all necessary data have been collected and reported comprehensively. | Indicators include missing data rates and consistency in reporting across different locations and periods. |
| Reliability | This reflects how consistently data is captured and reported over time and under varying conditions. | It considers the stability of reporting patterns, the robustness of tools used, and whether results can be reliably repeated. |

informing practical recommendations for implementing these technologies and guiding future research in low- and middle-income countries.

**Ensuring rigour and trustworthiness.** To maintain high standards in the qualitative part of the study, all findings will be evaluated using the JBI Checklist for Qualitative Research. We will enhance the credibility of our results by involving multiple reviewers in the analysis, with any disagreements resolved through team consensus. Transparency will be preserved through detailed documentation of our coding process and how themes were developed. Also, combining qualitative insights with the quantitative data (triangulation) will strengthen the overall validity and relevance of our findings.

## Conceptual framework for examining contextual and implementation factors affecting the efficacy of digital surveillance tools in LMICs

In LMICs, the success and effectiveness of digital surveillance tools heavily depend on various contextual and implementation factors [19] Gaining a clear understanding of these elements is essential for improving key data quality aspects like timeliness, accuracy, completeness, and reliability. This framework aims to systematically explore these factors to guide the deployment and expansion of digital surveillance systems across LMICs.

**Key contextual and implementation factors.**

1. Infrastructure:

    a. Digital Infrastructure: Reliable internet access, consistent electricity, and quality hardware form the backbone of digital surveillance systems. Without proper infrastructure, real-time data collection and timely reporting become challenging.

    b. Health Facility Infrastructure: The physical setups and organisational processes within health facilities, such as electronic health record systems and data management protocols, greatly influence how well digital tools work.

2. Workforce Capacity

    a. Training and Skills: The ability of health workers to use digital surveillance tools effectively relies greatly on their digital literacy and technical skills. Ongoing training and capacity building are essential.

    b. Human Resources: Sufficient staffing levels and dedicated personnel for data management ensure consistent and dependable data collection through digital platforms.

3. Data Security and Governance

    a. Privacy and Confidentiality: Protecting health data is critical. Insufficient data security measures could lead to breaches, which damage trust in digital systems.

    b. Regulatory Frameworks: Having clear policies on data use, sharing, and protection, along with proper enforcement, is essential for maintaining ethical and legal compliance.

4. Interoperability

    a. System Integration: Digital surveillance tools need to integrate smoothly with existing health information systems to enable smooth data flow and avoid redundancy.

    b. Standardisation: Following data standards makes it easier to exchange and compare data across different platforms and regions.

5. Sociocultural and community context

   a. Community trust: Evidence of local support or resistance; willingness to share data

   b. Engagement strategies: Use of community leaders or participatory design in tool deployment

   c. Misconceptions or stigma: Documentation of myths (e.g., digital tools causing harm or surveillance fears)

   d. Equity and inclusion: Representation of gender, ethnic, or vulnerable populations in rollout and uptake

**Conceptual framework diagram.** The diagram in Fig 1 illustrates how these contextual and implementation factors interrelate and collectively influence the success of digital surveillance systems in LMICs.

**Understanding the components.** Infrastructure: This is the essential base that supports everything else in the system. It includes both the physical components, like servers and networks, and organisational setups needed to effectively implement digital surveillance tools.

Digital Surveillance Tool Effectiveness: This measures how well the surveillance tools perform in tracking and gathering data. Their effectiveness depends heavily on the underlying infrastructure and is essential for reliable surveillance operations.

Workforce Capacity: Refers to the skilled personnel available to manage, operate, and maintain surveillance systems. Having a competent and adequately staffed team boosts the overall performance of these tools.

Data Security and Governance: Ensures that all data collected through surveillance is stored securely and managed responsibly. This includes implementing policies for data privacy, access control, and protection. Both the infrastructure and workforce play important roles in maintaining strong data security.

Interoperability: This describes how different surveillance systems can communicate and work together smoothly. When systems are interoperable, they share data smoothly, making the entire surveillance operation more efficient.

Sociocultural and community context: Our conceptual framework was thoughtfully expanded to incorporate a sociocultural dimension, emphasising the essential importance of community engagement, trust, and indigenous beliefs in influencing the acceptance and effectiveness of digital surveillance tools within low- and middle-income countries. By using the CFIR framework alongside socioecological implementation science models, we will systematically extract and integrate evidence on public perceptions, community participation, and the sociocultural factors, both barriers and facilitators, emphasised across each of the included studies.

**Relationship dynamics.** The arrows in the diagram show that Workforce Capacity and Data Security and Governance are both shaped by the underlying infrastructure and, in turn, influence the effectiveness of surveillance tools. Also, Data

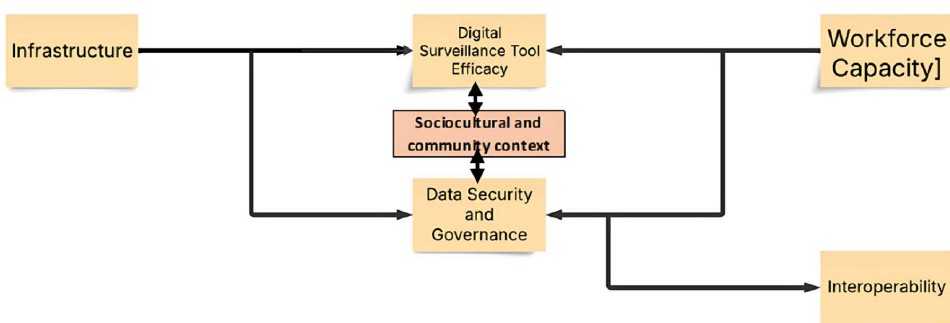

**Fig 1. Conceptual framework for examining contextual and implementation factors affecting the efficacy of digital surveillance tools in LMICs.**

Security and Governance are linked to Interoperability, suggesting that secure, well-managed systems are better at integrating with each other.

Overall, this framework emphasises how solid infrastructure supports effective digital surveillance, emphasises the importance of a skilled workforce and strong data policies, and stresses the need for systems to work together smoothly. All these elements are interconnected, influencing the overall capacity and reliability of surveillance activities.

This framework is designed to help us synthesise qualitative evidence in our systematic review. By grouping findings based on these factors, we can spot common barriers and facilitators across different settings. This approach offers valuable insights into what works well and what areas need more attention to ensure successful implementation.

## Quantitative analysis

**Approach to meta-analysis.** If the included studies are sufficiently homogenous, we will conduct meta-analyses using the random-effects model account for between-study variability. For studies that report comparable quantitative outcomes, we will compute the appropriate effect sizes (risk ratios, odds ratios, or mean differences).

Our goal with this meta-analysis is to bring together and compare findings from various studies on how digital surveillance tools perform in low- and middle-income countries (LMICS). We will focus on the primary and secondary outcome.

a. From each eligible study, we plan to extract the study details, outcome data and effect estimates.

To synthesise the data, we will use random-effects models (like the DerSimonian and Laird method) to handle variability across different studies caused by factors such as:

a. Different populations and settings (urban versus rural areas in LMICs).

b. Variations in surveillance tools, whether mobile apps, web-based platforms, etc.

c. Diverse outcome measurement methods.

Where applicable, we will analyse data using Risk Ratios (RRs) for binary outcomes (e.g., outbreak detection vs. non-detection), Standardised Mean Differences (SMDs) for continuous outcomes (e.g., reporting time in days), and Logit transformations for proportion data (such as rates of adoption or completeness).

Our meta-analyses will be run using R (packages like 'meta' and 'metafor') or Stata, depending on how the data looks. The results will be visualised through Forest plots showing pooled effect sizes. Funnel plots will be used to assess publication bias, while Meta-regression graphs will be used to explore effect moderators. This approach aims to provide a comprehensive understanding of how effective digital surveillance tools are across different LMIC contexts.

**Units of analysis.** The unit of analysis will vary depending on the level of measurements in each study. Studies reporting individual-level outcomes (clients, health workers) will be assessed at the individual level, while those measuring individual facility-level (hospital, clinics, or laboratories) outcomes will be analysed at the facility level. Finally, studies measuring digital surveillance programmes at national or sub-national levels will be analysed at the system level.

**Dealing with missing data.** It will be important to address missing data in some contexts. For example, if data is unclear, not fully reported, or absent from post-intervention reports despite being reported at baseline. The authors will deal with missing data using a combination of strategies. Where possible, we will contact the authors directly by email. Where possible, multiple imputation techniques will be used for continuous variables. We will conduct sensitivity analyses (excluding studies with substantial missing data) to evaluate the consistency of our findings. When missingness cannot be accounted for, we will use a narrative approach to explain its probable impact on the study findings. Finally, we will report all missing outcome data.

**Assessing heterogeneity.** We will use a combination of statistical and qualitative methods to assess heterogeneity. We will use common sense to determine whether the digital tools, participants, and outcomes can be combined. Also, we

will visually inspect graphs and forest plots for outliers, between-study differences, and overlapping confidence intervals. Cochran's Q test for heterogeneity (p < 0.10) and the F-test will also be used. For the former, p < 0.10 indicates significant heterogeneity, while the F test will quantify heterogeneity as low ($I^2 < 25\%$), moderate ($I^2 = 25–50\%$), or high ($I^2 > 50\%$). We will use subgroup (study design, region, digital tool type, surveillance level, and data quality indicator) analyses to explore the potential sources of heterogeneity. If possible (with an abundance of studies), we will explore the effect of study-level covariates on heterogeneity by conducting meta-regression.

**Assessment of reporting bias.** We will construct a funnel plot to detect small-study effects or selective publication. For meta-analyses of at least 10 studies, Egger's regression test will be used to detect publication bias. If there are fewer than ten studies, we will use the method outlined in the Cochrane Handbook. In the case that publication bias exists, a trim-and-fill method will be used to adjust for possible missing data and estimate the corrected effect size.

**Certainty of findings.** We will assess the certainty of evidence for each primary and secondary outcome using the GRADE approach, which considers five domains: risk of bias, inconsistency, indirectness, imprecision, and publication bias. The overall certainty will be rated as high, moderate, low, or very low.

Risk of Bias: We will evaluate each study using appropriate tools (e.g., ROBINS-I for non-randomized studies, and RoB 2 for randomized controlled trials). If studies show serious limitations, the certainty of evidence may be downgraded.

Inconsistency: Heterogeneity will be assessed (e.g., via $I^2$ statistic). Unexplained variability may lead to downgrading.

Indirectness: Downgrading will occur if populations, interventions, or outcomes differ substantially from those of interest.

Imprecision: Wide confidence intervals or small sample sizes may reduce certainty.

Publication Bias: Funnel plots and Egger's test (if ≥10 studies) will be used to assess bias in reporting.

Inconsistency: Heterogeneity will be assessed (e.g., via $I^2$ statistic). Unexplained variability may lead to downgrading.

Indirectness: Downgrading will occur if populations, interventions, or outcomes differ substantially from those of interest.

Imprecision: Wide confidence intervals or small sample sizes may reduce certainty.

Publication Bias: Funnel plots and Egger's test (if ≥10 studies) will be used to assess bias in reporting.

Findings will be summarised in GRADE Summary of Findings tables using GRADEpro GDT, showing the number of studies, effect estimates, certainty levels, and key comments.

Typically, randomised controlled trials are considered high-certainty evidence from the outset, whereas non-randomised studies generally begin with a lower certainty rating. The evidence's quality may be downgraded based on five main concerns: risk of bias, inconsistency, indirectness, imprecision, and publication bias. For instance, inconsistency is identified when the results vary substantially across studies, such as when the $I^2$ statistic exceeds 50%. Imprecision is usually evaluated by examining the width of confidence intervals or the sample size involved. On the other hand, non-randomised studies can be upgraded if they meet certain criteria. These include demonstrating a large effect size, such as risk ratios greater than 2 or less than 0.5 without major bias, a clear dose-response relationship, or when residual confounding factors are more likely to diminish rather than exaggerate the observed effect. In such cases, the strength of the association and the overall robustness of the evidence justify an increase in the certainty level. This approach offers a transparent and systematic method for assessing the quality of evidence, which is particularly valuable when integrating findings from a variety of study designs.

## Ethical considerations and declarations

Since this study is a review of already published and publicly available literature, it doesn't involve collecting new data from people. Therefore, it does not need formal ethical approval. However, we still follow ethical standards for research, such as being transparent in our reporting, avoiding plagiarism, and properly citing all sources. Any conflicts of interest among the authors will be openly declared. The authors confirm that they have no competing interests related to this

review. Currently, this study is not funded by any outside organisation. If we receive funding during the process, we will acknowledge it in future updates and publications.

### Status and timeline of the study

As of now, we are in the planning stages of the project. We are yet to commence the study.

Here's how we plan to proceed from 1st August 2025:

Database search and selecting studies: Months 1–3 (August to October 2025)

Data extraction and assessing quality: Month 4–5 (November 2025 to January 2026)

Analysing data and evaluating with GRADE: Month 6–7 (February to March 2026)

Writing up findings and peer review: Month 8 (April 2026)

Submitting the manuscript for publication: Month 9 (May 2026)

We expect to finish the review and submit the paper for peer review within nine months of starting the search process.

## Discussion

This review aims to bring together what we know about how the context and ways of putting digital surveillance tools into action affect their success in low- and middle-income countries. It also points out some important challenges and considerations along the way.

### Limitations of the study design

While we follow strict standards in our approach, there are some limitations to consider. For example, the studies vary a lot in their methods (like cross-sectional versus cohort studies), the populations they involve, and how the digital tools are used. This variety can make it harder to compare and combine results. Although we use an adapted version of the ROBINS-I tool to evaluate bias in different types of non-randomised studies, some issues, like confounding factors and differences in reporting, may still affect how reliable the overall findings are.

### Amendments and deviations in the protocol

Any major changes to our original plan, such as adjustments in what studies to include, what outcomes of interest, or how we analyse the results, will be documented and shared on PROSPERO. We will include details about when each change was made and why. If unexpected difficulties or practical issues threaten to undermine the review, we may decide to end it early after discussing with senior team members and stakeholders. In such cases, we will publish an explanation of what happened.

### Dissemination plans

We plan to publish the results in open-access, peer-reviewed journals and present them at international conferences focused on public health and digital health. To help put this knowledge into practice, we also aim to develop a policy brief for stakeholders in low- and middle-income countries. We will share summaries with digital health practitioners and implementers through webinars and workshops, and whenever possible, translate these summaries into local languages to make them more accessible and useful.

## Supporting information

**S1 Appendix.  Search strategy.**
(DOCX)

**S2 Appendix.  Data extraction tool.**
(XLSX)

**S3 Appendix. PRISMA-P checklist.**
(DOCX)

## Acknowledgments

Dr Victor Ayeni encouraged us even before we thought it was possible, and all through the protocol design.

## Author contributions

**Conceptualization:** Oluwatosin Olu-Abiodun, Olumide Abiodun.

**Methodology:** Aderinsola Faturoti, Davies Adeloye, Olumide Abiodun.

**Supervision:** Akindele Adebiyi.

**Writing – original draft:** Oluwatosin Olu-Abiodun, Aderinsola Faturoti, Akinmade Adepoju, Olumide Abiodun.

**Writing – review & editing:** Oluwatosin Olu-Abiodun, Aderinsola Faturoti, Akinmade Adepoju, Davies Adeloye, Akindele Adebiyi, Olumide Abiodun.

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
