## [Decision Letter · Decision Letter 0]

16 Jul 2025

PONE-D-25-30726
Effectiveness and Challenges of Digital Tools Implementation for Enhancing Infectious Disease Surveillance Data Quality in Low- and Middle-Income Countries: A Systematic Review Protocol
PLOS ONE

Dear Dr. Abiodun,

Thank you for submitting your manuscript to PLOS ONE. After careful consideration, we feel that it has merit but does not fully meet PLOS ONE’s publication criteria as it currently stands. Therefore, we invite you to submit a revised version of the manuscript that addresses the points raised during the review process.

We look forward to receiving your revised manuscript.

Kind regards,

Nishant Premnath Jaiswal, MBBS, PhD

Academic Editor

PLOS ONE

Journal Requirements:

Nil

4. Please amend your authorship list in your manuscript file to include all authors.

5. Please amend your list of authors on the manuscript to ensure that each author is linked to an affiliation. Authors’ affiliations should reflect the institution where the work was done (if authors moved subsequently, you can also list the new affiliation stating “current affiliation:….” as necessary).

6. Please ensure that you refer to Figure 1 in your text as, if accepted, production will need this reference to link the reader to the figure.

7. We note you have included a table to which you do not refer in the text of your manuscript. Please ensure that you refer to Table 1 in your text; if accepted, production will need this reference to link the reader to the Table.

8. Please remove all personal information, ensure that the data shared are in accordance with participant consent, and re-upload a fully anonymized data set.

Reviewers' comments:

Reviewer's Responses to Questions

**Comments to the Author**

1. Does the manuscript provide a valid rationale for the proposed study, with clearly identified and justified research questions?

Reviewer #1: Yes

Reviewer #2: Yes

2. Is the protocol technically sound and planned in a manner that will lead to a meaningful outcome and allow testing the stated hypotheses?

Reviewer #1: Partly

Reviewer #2: Yes

3. Is the methodology feasible and described in sufficient detail to allow the work to be replicable?

Reviewer #1: Yes

Reviewer #2: Yes

4. Have the authors described where all data underlying the findings will be made available when the study is complete?

Reviewer #1: No

Reviewer #2: Yes

5. Is the manuscript presented in an intelligible fashion and written in standard English?

Reviewer #1: Yes

Reviewer #2: Yes

6. Review Comments to the Author

You may also provide optional suggestions and comments to authors that they might find helpful in planning their study.

Reviewer #1: This is a systematic review protocol to collate the evidence regarding the Effectiveness and Challenges of Digital Tools Implementation for Enhancing Infectious Disease Surveillance Data Quality in Low- and Middle-Income Countries. The review protocol is well-written and I think the topic is quite relevant and novel. The rationale is well established given the challenges faced by traditional surveillance in LMICs and the growing adoption of digital tools. However, there are areas where clarity, flow, and organization could be improved for better readability and impact. Firstly, while the focus on LMICs is appropriate, the protocol would benefit from explicitly stating the criteria used to define LMICs (e.g., World Bank classification) and specifying which countries or regions are included. This will enhance reproducibility and transparency. Secondly, there is lot of repetition, avoid repeating ideas (e.g., importance of surveillance, LMIC challenges) across sections. Consolidate related information into a single, strong paragraph per theme.

Other concerns pointwise

Abstract:

1. The PROSPERO ID 1023840 is not traceable. Please check.

2. Language for sentence “we are interested in studies…….” should be changed to “types of studies will be included for better professional language.

3. ROBINS tool can be used for non-randomized studies, but for RCTs, ROB2 tool should be used.

Introduction:

1. References should be placed at appropriate places. For e.g., in the first paragraph, Nathan et al is placed inappropriately. Please cite reference for “The last two decades have been characterized by integrating digital technology into health care, which has led to several digital health platforms for strengthening infectious disease surveillance”.

2. The last paragraph in the introduction is just a repetition and has already been mentioned in the introduction, can be omitted.

Methods:

1. Although you broadly describe eligible study designs and populations, it would help to list explicit inclusion and exclusion criteria

2. A detailed search strategy for all databases should be provided.

3. The authors should clarify the rationale for selecting studies published only between January 2000 and April 2025. Explaining why this timeframe was chosen will help justify the scope of the review and ensure that relevant earlier or more recent studies are not inadvertently excluded.

4. The methodology for selection of studies should be explained explicitly. The authors have not mentioned the Ti/Ab screening followed by full text screening.

5. Mention how to tackle deduplication of citations, manually or through software or both.

6. The title “I. Qualitative Synthesis” seems misleading since the paragraph is about Quantitative synthesis.

7. You mention following PRISMA 2020, which is excellent. Consider specifying that the final review report will include a PRISMA flow diagram to show the study selection process

8. The whole section of “Quantitative analysis Approach to Meta-analysis” is a repetition described previously also. Please clarify.

9. Your plan for thematic analysis and mixed-methods integration is strong. Consider specifying the software or frameworks.

Overall, minor grammatical issues such as inconsistent capitalization (e.g., “Infrastructural”), missing commas, and awkward phrasing appear throughout. Maintain consistent terminology (e.g., “digital surveillance tools” vs. “digital platforms”). Break long paragraphs into smaller ones for readability, especially in the Background section.

Reviewer #2: The protocol has been written in a very appropriate way including all the steps needed for conducting a systematic review. However, there are some minor concerns which author can take care of:

a) Please provide the reference s for the tools/ guidelines to be used in developing the systematic reviews such as PRISMA 2020, JBI etc.

b) Qualitative and Quantitative section needs to be updated. In qualitative sysnthesis author talked about meta analysis of studies having same methodology, which is part of the quantitative synthesis.

c) There is a need of data extraction for to be added in supplementary section having all details pertaining to the data needed for qualitative and quantitative synthesis.

d) Author could check for the word limit of each of the section according to the journal requirement

7. PLOS authors have the option to publish the peer review history of their article (what does this mean?). If published, this will include your full peer review and any attached files.

Reviewer #1: **Yes: **Meenakshi Sachdeva

Reviewer #2: No

---

## [Author Response · Author response to Decision Letter 1]

22 Jul 2025

Manuscript ID:

Response to Reviewers

We sincerely appreciate the reviewers for their careful evaluation and insightful feedback on our manuscript. Their dedicated effort in offering thoughtful comments has greatly enhanced the clarity, quality, and depth of our work. In this document, we have systematically addressed each review point. Reviewer comments are followed by our responses written in plain red text. When relevant, we have specified the particular changes made to the manuscript and indicated the exact locations of these modifications, such as section headings. We believe that the revisions and clarifications presented herein sufficiently resolve the concerns raised and contribute positively to the overall impact of the manuscript within the field.

Reviewer #1:

This is a systematic review protocol to collate the evidence regarding the Effectiveness and Challenges of Digital Tools Implementation for Enhancing Infectious Disease Surveillance Data Quality in Low- and Middle-Income Countries. The review protocol is well-written and I think the topic is quite relevant and novel. The rationale is well established given the challenges faced by traditional surveillance in LMICs and the growing adoption of digital tools.

Thank you for the kind comments

However, there are areas where clarity, flow, and organisation could be improved for better readability and impact.

Firstly, while the focus on LMICs is appropriate, the protocol would benefit from explicitly stating the criteria used to define LMICs (e.g., World Bank classification) and specifying which countries or regions are included. This will enhance reproducibility and transparency.

LMICs are defined according to the World Bank classification (countries with a gross national income (GNI) per capita of $13,845 or less) categorized using the Atlas method (World Bank, 2024).

Secondly, there is lot of repetition, avoid repeating ideas (e.g., importance of surveillance, LMIC challenges) across sections. Consolidate related information into a single, strong paragraph per theme.

The repetitions have been comprehensively addressed.

Other concerns pointwise

Abstract:

1. The PROSPERO ID 1023840 is not traceable. Please check.

I have done the necessary things in PROSPERO, and it should be active by the time of this review.

2. Language for sentence “we are interested in studies…….” should be changed to “types of studies will be included for better professional language.

The language has been improved to sound more professional

3. ROBINS tool can be used for non-randomized studies, but for RCTs, ROB2 tool should be used.

The quality of each study will be assessed using ROBINS-I for non-randomized studies and ROB2 for randomized controlled trials

Introduction:

1. References should be placed at appropriate places. For e.g., in the first paragraph, Nathan et al is placed inappropriately. Please cite reference for “The last two decades have been characterized by integrating digital technology into health care, which has led to several digital health platforms for strengthening infectious disease surveillance”.

WHO, 2021. The reference, Nathan et al, has been place appropriately

2. The last paragraph in the introduction is just a repetition and has already been mentioned in the introduction, can be omitted.

This paragraph has been removed.

Methods:

1. Although you broadly describe eligible study designs and populations, it would help to list explicit inclusion and exclusion criteria

Inclusion and Exclusion Criteria

To ensure pertinence, the following criteria were implemented:

Inclusion Criteria

a. Studies that assess the effectiveness of digital surveillance tools in infectious disease surveillance

b. Studies that detail data quality indicators (accuracy, timeliness, completeness, reliability)

c. Studies conducted in LMICs as categorized by the World Bank

d. Randomized controlled trials, observational studies, qualitative studies, and systematic reviews

e. Peer-reviewed articles published from 2000 to 2025

Exclusion Criteria

a. Studies not connected to infectious disease surveillance

b. Studies concentrating solely on non-communicable diseases (NCDs)

c. Studies lacking measurable data quality indicators

d. Articles published in languages other than English (unless an English translation is provided)

e. Conference abstracts, commentaries, and opinion pieces without empirical data

2. A detailed search strategy for all databases should be provided.

Detailed search strategy is provided in Appendix 1

3. The authors should clarify the rationale for selecting studies published only between January 2000 and April 2025. Explaining why this timeframe was chosen will help justify the scope of the review and ensure that relevant earlier or more recent studies are not inadvertently excluded.

This is because many of the major advances in digital health technologies, such as mobile health, electronic monitoring platforms, and spatial data systems, began to grow and become widespread around that time, especially in low- and middle-income countries. Also, global efforts to improve disease surveillance, like the World Health Organization’s revised framework called the Integrated Disease Surveillance and Response (IDSR), have mostly been developed after 2000 (WHO, 2010). Because of this, studies published before 2000 are less likely to include information about the modern tools, infrastructure, and systems used for digital disease monitoring in these settings (Akanbi et al., 2012).

4. The methodology for selection of studies should be explained explicitly. The authors have not mentioned the Ti/Ab screening followed by full text screening.

Study selection will be conducted in two stages: first, a screening of titles and abstracts to identify potentially eligible studies; and second, a full-text screening of those studies retained after the first stage.

5. Mention how to tackle deduplication of citations, manually or through software or both.

Duplicate citations will be automatically detected and eliminated through the deduplication feature in EndNote X8.2. Following this, a manual review will be conducted to verify that no duplicates remain and to confirm the thoroughness of the process.

6. The title “I. Qualitative Synthesis” seems misleading since the paragraph is about Quantitative synthesis.

Indeed, there was an error there. We have corrected this.

7. You mention following PRISMA 2020, which is excellent. Consider specifying that the final review report will include a PRISMA flow diagram to show the study selection process

The final review report will incorporate a PRISMA 2020 flow diagram, providing a clear visual overview of the study selection process. This diagram will illustrate the number of records identified, screened, excluded, and included at each respective stage to assist better understanding.

8. The whole section of “Quantitative analysis Approach to Meta-analysis” is a repetition described previously also. Please clarify.

This section has been revised, and the repetitions have been eliminated

9. Your plan for thematic analysis and mixed-methods integration is strong. Consider specifying the software or frameworks.

NVivo will be used for qualitative analysis. For studies with mixed methods, a convergent synthesis approach will be used to integrate qualitative findings with quantitative results

Overall, minor grammatical issues such as inconsistent capitalization (e.g., “Infrastructural”), missing commas, and awkward phrasing appear throughout. Maintain consistent terminology (e.g., “digital surveillance tools” vs. “digital platforms”). Break long paragraphs into smaller ones for readability, especially in the Background section.

These issues have been carefully addressed

Reviewer #2:

The protocol has been written in a very appropriate way including all the steps needed for conducting a systematic review.

Thank you

However, there are some minor concerns which author can take care of:

a) Please provide the reference s for the tools/ guidelines to be used in developing the systematic reviews such as PRISMA 2020, JBI etc.

The appropriate references have now been included.

b) Qualitative and Quantitative section needs to be updated. In qualitative sysnthesis author talked about meta analysis of studies having same methodology, which is part of the quantitative synthesis.

This has been corrected.

c) There is a need of data extraction form to be added in supplementary section having all details pertaining to the data needed for qualitative and quantitative synthesis.

The proposed data extraction form is provided in Appendix 2

d) Author could check for the word limit of each of the section according to the journal requirement

The word limits have been reviewed, and we have made reasonable efforts to comply with them.

---

## [Decision Letter · Decision Letter 1]

8 Aug 2025

Effectiveness and Challenges of Digital Tools Implementation for Enhancing Infectious Disease Surveillance Data Quality in Low- and Middle-Income Countries: A Systematic Review Protocol

PONE-D-25-30726R1

Dear Dr. Abiodun,

We’re pleased to inform you that your manuscript has been judged scientifically suitable for publication and will be formally accepted for publication once it meets all outstanding technical requirements.

Kind regards,

Pasyodun Koralage Buddhika Mahesh

Academic Editor

PLOS ONE

Additional Editor Comments (optional):

Reviewers' comments:

Reviewer's Responses to Questions

**Comments to the Author**

1. Does the manuscript provide a valid rationale for the proposed study, with clearly identified and justified research questions?

Reviewer #1: Yes

Reviewer #2: Yes

2. Is the protocol technically sound and planned in a manner that will lead to a meaningful outcome and allow testing the stated hypotheses?

Reviewer #1: Yes

Reviewer #2: Yes

3. Is the methodology feasible and described in sufficient detail to allow the work to be replicable?

Reviewer #1: Yes

Reviewer #2: Yes

4. Have the authors described where all data underlying the findings will be made available when the study is complete?

Reviewer #1: Yes

Reviewer #2: Yes

5. Is the manuscript presented in an intelligible fashion and written in standard English?

Reviewer #1: Yes

Reviewer #2: Yes

6. Review Comments to the Author

You may also provide optional suggestions and comments to authors that they might find helpful in planning their study.

Reviewer #1: The authors have done the necessary changes and incorporated the suggestions. The manuscript is now fit for publication.

Reviewer #2: None. The comments have been addressed appropriately. Data extraction sheet has been added and the protocol is now well formulated.

7. PLOS authors have the option to publish the peer review history of their article (what does this mean?). If published, this will include your full peer review and any attached files.

Reviewer #1: **Yes: **Dr Meenakshi Sachdeva

Reviewer #2: No

---

## [Editor Report · Acceptance letter]

PONE-D-25-30726R1

PLOS ONE

Dear Dr. Abiodun,

I'm pleased to inform you that your manuscri004
pt has been deemed suitable for publication in PLOS ONE. Congratulations! Your manuscript is now being handed over to our production team.

Kind regards,

on behalf of

Dr. Pasyodun Koralage Buddhika Mahesh

Academic Editor

PLOS ONE